# Boundaries of Belonging: Theorizing Black African Migrant Experiences in Australia

**DOI:** 10.3390/ijerph18010038

**Published:** 2020-12-23

**Authors:** Kathomi Gatwiri, Leticia Anderson

**Affiliations:** School of Arts and Social Science, Southern Cross University, Gold Coast, QLD 4225, Australia; leticia.anderson@scu.edu.au

**Keywords:** African diaspora, migration, Australia, belonging, politics of belonging, bordering, racism

## Abstract

As nationalist ideologies intensify in Australia, so do the experiences of ‘everyday racism’ and exclusion for Black African immigrants. In this article, we utilize critical theories and engage with colonial histories to contextualize Afrodiasporic experiences in Australia, arguing that the conditional acceptance of Black bodies within Australian spaces is contingent upon the status quo of the white hegemony. The tropes and discourses that render the bodies of Black African migrants simultaneously invisible and hyper-visible indicate that immigration is not only a movement of bodies, but also a phenomenon solidly tied to global inequality, power, and the abjection of blackness. Drawing on critical race perspectives and theories of belonging, we highlight through use of literature how Black Africans in Australia are constructed as ‘perpetual strangers’. As moral panics and discourses of hyper-criminality are summoned, the bordering processes are also simultaneously co-opted to reinforce scrutiny and securitization, with significant implications for social cohesion, belonging and public health.

## 1. Introduction

Social anxiety in regards to immigration is not a new phenomenon in Australia, a settler-colonial country that has a long history of exclusion and marginalization towards Black people and other people of colour. Despite this historic context, the promotion of migration over successive decades in the latter part of the twentieth century led to significant demographic change as the nation became highly culturally and racially diverse. Amidst this increasing diversity, concerns about crises of immigration were publicly summoned from the late 1990s by political figures whose fantasies of the restoration of a white Australia were eventually coopted by mainstream conservative political actors and commodified for political leverage [1]. Contemporary cultural and political discourses on immigration in Australia continue to indicate how power is used to discipline and control bodies that are deemed dangerous, different, deviant, and unassimilable. As is well documented in the literature, the Black African body is typically constructed as ‘other’ in various ways within the Australian context. By popularizing the idea of strange and alien bodies as a signifier for protecting national boundaries, Australia can strive to maintain a state of white fantasy mediated by ideologies of securitization [2]. As an example, in the wake of mediatized ‘moral panics’ regarding the alleged criminality of African migrants, claims about their inability to ‘integrate’ successfully into Australia society were connected to reductions in humanitarian resettlement from African nations [3]. Subsequent cuts to migration in the lead-up to the 2019 Federal Election have similarly been decried by some as racialised anti-immigrant dog whistling. As the rhetoric of ‘border-control’ gains momentum, anxieties about out-of-place bodies reach new levels of amplification and thus require careful theorization.

While much research has been conducted on the lived experiences of African migrants, this has predominantly focused on the Global North particularly in North America and Europe. As Mapedzahama and Kwansah-Aidoo have pointed out, much of the literature on African, diasporic existence has concentrated on the forced trans-Atlantic migration of Black Africans into America during slavery and on refugeeship [4]. In Australia, research on African migrants is increasing but has to date focused primarily on refugees. There is a tendency towards over-researching South Sudanese communities, often driven by efforts to contend with recurrent media stigmatization of these communities [3]. Research about other African diasporic experiences in Australia, such as skilled African migrants or second/third generation migrants, and especially research conducted by researchers who have personal experience of migration to Australia from African nations, is still an emerging field [1]. There is therefore a need for theoretical writing on alternate African diasporic experiences and identities and the changing nature of such identities, especially writing that is sensitive to and generated through Afrocentric perspectives to avoid monolithic arguments that are reductionist towards the experiences of diasporic Africans. Udah and Singh found in their research, that regardless of their reasons for migration and backgrounds, Black Africans migrants in Australia frequently experienced reductive assumptions that singularized their identities [5]. This research reported that these experiences negatively impacted their health, wellbeing and sense of belonging. Extending the theorization of blackness to include multiple complexities of Black identities and the resulting experiences, is necessitated as these challenges the dominance of the “single story” [6] about Black Africans. 

This paper provides theory-based arguments elucidation how mainstream immigration attitudes in Australia impact upon Black African migrants through the mediums of mediatized moral panics about Black criminality and acts of everyday racism that construct and enforce their ‘strangerhood’. It explains why research on the movement of Black bodies across boundaries and geographies must incorporate histories of colonialism, racism, marginalization and their impact on the lived experience of Black people in predominantly white countries. By employing critical race perspectives to understand the subjective experiences of Black African migrants in Australia, we argue that migration is more than just movement of bodies, it is a phenomenon solidly tied to global inequality, power, and the abjection of Black bodies. In this paper, and similar to other Afrodiasporic scholars [7,8,9], we use ‘Black and Blackness’ to refer only to ‘Afro-Blackness’ or the blackness that is experientially embodied by people identified as ‘Black Africans’. We do this while also acknowledging that not all Black people are of African descent, and not all Africans are Black. While we are careful not to homogenize the experiences of Black Africans in Australia, we acknowledge that bodies that are visibly marked as ‘Black’ and ‘African’ can share some similarities in their experience.

## 2. Contextualizing Migration of Africans in Australia

African migration to Australia has a long history, comprising different waves of migration, under various circumstances, but large-scale immigration flows are a relatively recent phenomenon. Prior to and directly following the Second World War, the majority of African migrants in Australia were white South Africans. However, this majority began to decrease from the 1960s onwards, due to the dismantling of racial discrimination in immigration policies. The Africa-born population in Australia grew steadily during the late twentieth century, with migrants eventually drawn from almost every country in Africa and increasingly of Black African heritage [10]. Between the 2001 and 2011 Australian Censuses, the number of people born in Sub-Saharan African countries living in Australia doubled, and by 2016, the total for all Africa-born in Australia had increased to 388,683, or approximately 1.7% of the total population [11]. Although a small proportion of the overall population, the increase of Black African diaspora communities in Australia has led to high levels of public and media scrutiny, through the deployment of racialized tropes that portray them as members of monolithic social categories [12].

People wishing to permanently migrate to Australia currently enter via one of two visa ‘streams’: the Humanitarian program for resettlement of refugees, and the Migration program, for family-based and skilled work migration. This is in addition to a variety of temporary migration programs, including for skilled workers and international students [13]. The intake of African-born migrants via the Humanitarian resettlement program was substantially reduced in the mid-2000s, and successive governments have aimed at reducing family reunion or ‘chain’ migration for several decades. Black African migration to Australia is therefore increasingly occurring via the skilled migration stream than in other pathways [11,14]. Skilled migration has been championed by successive governments as it directly promotes that notion that, ‘(skilled) migrants would add to the nation’s economic prosperity without threatening social cohesion’ [14], p. 21. 

In sectors such as healthcare where there is a high demand for migrant workers with relevant Bachelor- or postgraduate-level qualifications, there have been expanding opportunities for permanent migration to Australia for skilled African workers and their families [15,16]. However, despite the increasing size and diversity of African diaspora communities in Australia, Gatwiri and Anderson argue that the social, cultural and political positioning of Black Africans in Australia remains dominated by deficit discourses. This ‘can lead to all Black African migrants being synonymized as refugees, assumed to have experienced trauma, and stereotyped as lacking in education, professional expertise, and English proficiency’ [17], p. 2. Udah and Singh posit that such deficit discourses pathologize and inferiorize Black Africans ‘problematizing them as lacking in something [18], p. 37 which complicates efforts to achieve a sense of belonging in Australia.

## 3. Theorizing Belonging in Australia

To understand subjective migratory experiences, we must critically theorize how bordering practices are summoned when colonized and racialized bodies cross international boundaries and how that positioning impacts upon their ability to belong in the new country. We frame belonging through Nira Yuval-Davis’s (2006) theorization which locates belonging as an ‘emotional attachment, [and] about feeling at “home” [19], or as Ramon Spaijj has elaborated, as having ‘a sense of being part of the social fabric’ [20], p. 304. As belonging becomes politicized, it morphs into concept that is mediated by power [21]. Yuval-Davis, Wemyss and Cassidy suggest that belonging can be politicized, for example, when it is linked to the bordering and securitization processes through which national polities construct ‘views on who has a right to share the[ir] home and who does not belong there’, that is, establishing a nationalized collective ‘us’ that is separate to/from ‘them’ [22], p. 7. As borders function symbolically and metaphorically as tools of geographical and colonial separation, the bordering processes that are summoned produce state-sanctioned ‘expressions of sovereignty’ that can be utilized to police those who have the right to cross certain borders and to join ‘us’, and those who do not [21], p. 73.

The borders established between ourselves and *Others* therefore operate to protect hegemonies, and privilege some groups over others. Whilst “all bordering processes are a combination of ordering and othering… [that] differentiate ‘us’ and ‘them’”, [22], p. 5 categories dividing people into identity-based groups—especially those based on race—frequently have a history that is intimately connected with colonialism and the establishment of contemporary global hegemonies. Critical perspectives on the bordering practices of colonialism/post-colonialism question the identities, knowledges and traditional (Western) scholarship that work to privilege the White Western body over the non-Western body [22,23,24]. This body of knowledge also questions what we are socialized to think of as ‘natural boundaries/borders’ while interrogating the processes of border formation. In doing so, it draws strongly upon the complex ‘border-crossings’ and multifaceted experiences of groups of people who have been ‘othered’ through the processes of colonial bordering, including migrants, refugees, and Indigenous peoples.

Critical perspectives on coloniality also emphasize that borders are not only drawn on the land, but in our minds and bodies as well. In her landmark work Borderlands/La Frontera, for example, scholar and author Gloria Anzaldúa explored the significance of ethnicity, gender and sexuality and their intersections with various political and cultural boundaries through the creation of a complex ‘borderlands biography’ that illuminated the layering of contested meanings within the personal as well as the physical borderlands she inhabited [25]. As Leanne Weber observes, intersectional ‘markers of difference and exclusion’ produce hierarchical ‘categories’ of citizenship.

*Markers of difference and exclusion are often associated with hierarchies of citizenship. These hierarchies, whether legally defined or socially produced through the structural effects of colonization, gender, race, class or nationality, effectively sort populations into categories marked (to varying degrees and in particular contexts) as either full or partial citizens*.[21], p. 73

In the Australian context, asymmetry in regards to the rights of particular groups to police and maintain boundaries of belonging frequently plays out in relation to racialized discourses about Australian identity [21]. Ghassan Hage’s theorisation of ‘governmental belonging’ has increasingly being deployed in order to explain the particular boundary work that is enacted by those who able to assert ‘proprietorial’ or ‘governmental’ rights to belonging and Australian identity within a variety of contexts [20]. He argued that through greater congruence with the dominant (white) national culture, some are better positioned than others to accumulate national and social capital and if suitably motivated, can easily accumulate and hold claims to what he termed ‘governmental belonging’. It is ‘governmental belonging’ that affords some the position of cultural and political dominance it’ [2]. Anderson and colleagues have also argued that those that are mostly afforded governmental belonging are Australians of white settler-colonial heritage [who] are…uniquely articulated as ‘locals’ … enabling them to simultaneously reject the prior claims of possession by Indigenous Australians and any subsequent claims to belonging’ [26], p. 27.

These rights are asymmetrical, as Yuval-Davis alluded [19], in that, the ability to grant or deny the claims of belonging within the Australian socio-political milieu is ‘claimed by those who are in a dominant position and can lead to minoritised individuals or groups being silenced and positioned as “other”’ [20], p. 305. Claims to belonging by those who are ‘othered’ within this environment are heavily constrained and policed. Cultural difference must be carefully managed within Australian civic spaces, in a way that does not destabilize white racial comfort or proprietary claims to belonging, otherwise even tentative claims to belonging within these spaces can be denied or even retracted. Anderson and Gatwiri have probed, for example, the way in which Black women are harshly policed and disciplined by the mainstream Australian media if they provoke racial discomfort. The punishment of racial transgressors is often ‘deployed in a brutal, complex array of socially-sanctioned patterns which include penalization, retaliation, and ostracization’, as a way of delineating the boundary lines they must toe while simultaneously reminding them of the power of white hegemony to silence [27]. As such, acceptance of racialized bodies and voices within Australian public spaces is a process of invoking conditional belonging, and is always contingent upon the granting of such ‘rights’ by hegemonic groups. As Kwansah-Aidoo and Mapedzahama highlight, establishing a sense of ‘belonging-as-negotiation’ [28], p. 110 requires that individuals may have to come to terms with the likelihood that they will ‘occupy, what Gloria Anzaldua’s refers to as “the borderlands”; that is, they are in-between places and will be juggling cultures” [25,28]. A legitimate strategy for others is the contemplation of/or decision to ‘return home’ [28]. That is, to relocate back to their African country of origin, in the search for a more dignified sense of belonging, where they are not relegated to the margins.

Tying together decolonial concerns and the historical complexities of national, cultural, and personal borders enables understanding of emerging migrant experiences of border-crossing and belonging. When Black Africans migrate to Western and settler-colonial societies, the relegation of individuals from a wide variety of cultural, ethnic and national backgrounds to the singular category of ‘African’ strips them of both individual and national identities. It can make their Black bodies hyper-visible, but simultaneously invisible, in the sense that their presence, voice, and contributions may be repeatedly downplayed, ignored, or silenced [4]. These complexities of Australia’s settler-colonial history influence and shape contemporary projects of border control.

## 4. Bordering Projects and Practices in Australia

Australia is a settler-colonial society and prior to colonization, it was home to several hundred distinct Indigenous nations, who have never ceded sovereignty. The failure to acknowledge Indigenous sovereignty and continuous denials of the extent and ongoing negative impacts of colonization continues to cast a pall upon contemporary Australian society [29]. In addition to the legacies of the dispossession of Indigenous peoples, the development of Australian national identity was characterized by a steadfast exclusion of ‘others’ based on race. A range of policies and laws collectively known as the White Australia Policy were implemented to restrict the ability of people of non-Anglo heritage to enter, work and live freely in Australia. These policies were effective in engineering a racially and culturally homogenous society, but their progressive dismantling after World War II led to significant demographic change in the latter part of the twentieth century [30]. Almost a quarter of Australians now have non-European heritage, compared to the extremely small minority during the period of the White Australia Policy [31].

Despite the multicultural reality of contemporary Australian society, ‘dominant narratives of “Australianness” and belongingness continue to revolve around the centrality of whiteness and “Anglo-Celtic” heritage to Australian identity’ [26], p. 24. Gwenda Tavan has argued that the political success of immigration was founded in an implicit ‘bargain’ that immigrants would ‘give up their foreignness’ enabling Australian institutions and dominant culture to remain unchanged by outside influence. This means that ‘while immigration might be an economic necessity, it would remain tangential to Australian national identity’ [32], pp. 159–160. Reflecting this proposition, although Australians of non-European heritage have become a much larger proportion of the population, they remain underrepresented in positions of leadership within government and industry, and within key cultural industries [31].

Recurrent anxieties around demographic change and perceived threats to ‘border-control’ evidence the continuity of white hegemony, reflect the unresolved legacies of Australia’s colonial past and present, and are linked to governmental distancing from multiculturalist policies. In 2007, for example, the Department of Immigration and Multicultural Affairs was renamed the Department of Immigration and Citizenship, which subsequently became the Department of Immigration and Border Protection, and by 2017 had been absorbed into Home Affairs, a new super-department [33]. During the same period, Australian policies towards migrants and asylum-seekers became increasingly restrictive and punitive. These developments signal a transition away from political and social endorsement of multiculturalism and a reinvigoration of racialized approaches to immigration, exacerbated by the use of ‘border control’ as a wedge issue for political benefit [34].

Contradictory evidence regarding mainstream acceptance of cultural diversity repeatedly resurfaces in large-scale social attitude surveys about national identity and cultural cohesiveness. On the one hand, such surveys often indicate strong affirmation that migration has been good for Australia, but simultaneously reveal continued support for assimilation. As an example, in research commissioned by a public broadcaster, 80% of respondents agreed it was good for a society to be made up of people from different cultural backgrounds, yet 49% implicitly endorsed assimilation, reporting agreement with the statement that ‘people from racial, ethnic, cultural and religious minority groups should behave more like mainstream Australians’ [35], p. 6. Whilst ‘assimilation’ is no longer government policy, it prevails through microagressive practices and exists as a tacit presumption of the ‘bargain’ on immigration. This means complete assimilation is the condition of successful integration of migrants and Indigenous people and is the ultimate measure of good multiculturalism. Contradictory tensions towards acceptance and exclusion of migrants are particularly fraught for people who cannot visibly ‘assimilate’. Migrants and refugees who are identified or identify as Black Africans ‘frequently report higher levels of discrimination and prejudice than other groups in Australian society’ [26], p. 25.

As globalization becomes normalized and the certainties of borders (physical and symbolic) become blurred, xenophobic nationalism can be embraced by dominant social groups as a way to assert and defend their hegemonies. As nationalist ideologies intensify in Australia, experiences of racism and exclusion also intensify for people of colour. In a large study conducted in 2016, almost a third of young Australians reported experiencing unfair treatment or discrimination based on their race [36]. A recent study on discourses of national identity in Australian schools found that ‘implicit white normativity’ still acts to ‘reinforce a white Anglo identity as the presumed “core” of Australian national identity… [this] condones the normative assumption that “real” Australians are white people and others with racially marked bodies and culturally “different” identities originate from elsewhere’ [37], p. 142. Young people of colour are particularly overrepresented in out-of-home care and in custody; it can be argued that ‘the criminal justice system has increasingly become a tool for substituting direct racial discrimination with less overt practices that still have discriminatory and exclusionary effects’ [38], p. 528. Given the historic positioning of whiteness as central to Australian identity and the escalating moral panics that demonize Black African migrants, achievement of a sense of belonging and acceptance within new communities can be complicated or even thwarted [1]. Kwansah-Aidoo and Mapedzahama have described this as relegating Black Africans to the status ‘of being a perpetual stranger who does not belong’ [28], p. 97.

## 5. Contextualizing the Black African Immigrant Experience in Australia

New understandings of difference and race have enabled the expansion of research on experiences of discrimination and racism in Australian society. Much of the recent re-search in this field is being conducted by or in association with academics who them-selves are part of African diaspora communities and/or are actively utilizing Afrocentric approaches to their work, and so we draw where possible upon this emerging body of research. Whilst being mindful of the need to avoid the pitfalls of reductionist generalizations, we observe some general trends and key themes and contextualize and theorize the experiences canvassed within this literature within two broad themes: [1] racialized criminality and moral panics and [2] perpetual strangerhood.

### 5.1. Racialized Criminality and Moral Panics

As Stuart Hall contended, representation is a process of producing meaning, which then shapes and constructs reality [39]. The politicization and commodification of racism frames Black Africans through a routined silencing of African ontological and epistemological experiences. These shared meanings/misunderstandings of blackness and Africanness are translated through language to operationalize representational discourses that justify ‘moral panics’, or ‘a turbulent and exaggerated response to a putative social problem’ that feeds off and disseminates ‘popular demonologies’ at a variety of interconnected local, national and international levels [40]. In reviewing the proliferation of literature invoking the concept of ‘moral panics’, Falkof writes that ‘a heightened sense of fear is a standard feature of our times’, yet simultaneously, ‘often the causes of our anxiety are invisible, creating the sense that we are at the mercy of a global system that we do not fully understand and cannot hope to influence’ [41], p. 233. In such a context, moral panics flourish because they effectively obscure meaningful critical analysis of the validity of these panics and instead allow for a projection of those fears onto minoritized others who can easily be scapegoated through these events.

Racialized moral panics have functioned in Australia as a key way in which the borders of belonging have been policed and patrolled. In the past two decades there has, for example, been a succession of moral panics about criminality and violence associated with African migrants [12]. Most recently, the resurfacing of this moral panic in mid-2018 led to the hashtag #AfricanGangs trending in social and popular media. Conservative political and media figures utilized a range of inflammatory and inaccurate statements about the supposed criminality of African migrants, with a focus on Sudanese refugee communities, to achieve political mileage. The city of Melbourne was positioned as a ‘terror zone’ and sensationalized news coverage fostered the perception that violence perpetrated by so-called ‘African gangs’ was so widespread that it warranted deportations [8]. In reality, the concerns about the supposed inherent and rising criminality of African migrants have been disproportionate and proven inaccurate [8].

Widespread anxieties subside once the focus of media shifts to alternative targets. However, the repeated re-emergence of moral panic in regards to Black criminality and violence is indicative of the power of anti-Black rhetorics in community and media representations. As Falkof reminds us, ‘certain folk devils or types of deviance are so potent that they reappear repeatedly even though no real proof is produced of the danger they are thought to pose’ [41], p. 232. Black African people are therefore represented as having fixed identities within white dominant societies, and according to common representational tropes, they cannot be good; they are dangerous and to be feared. Majavu has demonstrated that tropes and discourses about Africanness in Australia ‘draw from the global racialized archive of information about Negroes’ that cast them as ‘inherently dangerous, and thus in need of civilizing by white society’ [8], p. 35. This discourse functions as justification for over policing ‘Africans’ in Australia.

In their research with young people from South Sudanese backgrounds, Weber has documented how following stigmatization of this community, ‘enforcement officers on public transport and private security guards’ also began to police their movements [21], p. 79. Weber wrote that, ‘*police participate in … the politics of belonging when their [mis]treatment of migrant groups conveys messages about belonging to the wider population*’ [21]. Such experiences of ‘unfair targeting’ are interpreted as placing Black Africans ‘outside the boundaries of “secure belonging”’ [21], p. 83. Racialized anxieties fueled through such moral panics also have potential to impact on anyone who was identified (correctly or incorrectly) as Black and/or African through an increase of ‘everyday racism’ experiences [42]. Such personal experiences of hyper-scrutinization and violent policing, or legitimate fears about these practices, can also impact upon parenting practices and intergenerational relations within migrant communities [17].

Over policing of Black communities in Australia is not a new phenomenon, given the settler-colonial context outlined above. Cunneen argues that many of the characteristics associated with high rates of crime and incarceration for racialized peoples in countries such as Australia are ‘long-term outcomes of colonial policies’, with criminal justice processes working as ‘a way of “weeding out” those who fail the test of (White) social conformity’ [38]. Windle argues that the history of racism in Australia as well as the dominant narratives about the ‘gang-culture’ of Black people inform fear discourses directed towards Africans in Australia. They state:

*Racialisation of African refugees in the Australian media appears to find its proximate source in the activation of race as an explanatory category amongst police, giving license to a xenophobic minority. This activation draws on the history of racism in Australia, on wider colonial narratives about primitive Africa, on the perennial discourse of dangerous youth, and even on fears about American cultural imperialism (in the form of black “gang culture”). As with Indigenous Australians, the dominant frame is one of underlying societal risk*.[12], p. 563

Over-surveillance and policing of Black people indicates that they remain overwhelmingly constructed as ‘perpetual suspects’ or ‘persons of interest’. Labelling the Black body through these criminalized lenses functions as a key tool of ‘othering’. In this context, the Black African body remains political as it occupies a contested space. The confluence of blackness and criminality therefore functions to position them as troublemakers and a threat to safety in communities, and to preclude other aspects of their identities from being recognised and celebrated [8].

Benier et al.’s research with young people of South Sudanese backgrounds demonstrated the frustration participants felt with the limits of Australian multiculturalism. Any achievements by people within their communities, they observed, were lauded as evidence of their ‘Australianness’, whereas any ‘wrongdoing’ was ‘rarely associated with Australian culture, policies, institutions, or systemic barriers to social inclusion…instead, it was attributed to the “Africanness” of perpetrators’ [43], p. 32. In his analysis of media representations, Windle similarly found that there was a ‘disassociation of crime from “Australianess”’, with criminality repeatedly attributed to ‘Africans and other “outgroups” who need to “integrate” [12]’. As Falkof has pointed out, ‘the morality of the moral panic can cement the imagined community…when something outside of us is bad or evil or dangerous it may allow us to create positive collective identities by defining ourselves in distinction to the people or circumstances that imperil the stability of our moral worlds’ [41], p. 231. By projecting criminality onto the ‘Africanness’ of migrant communities and individuals, ‘mainstream’ Australians can be shielded from the need to grapple with the reality of racism and its impacts.

Although a moral panic by definition is subject to ‘ebbs and flows’, and the anxiety produced through such episodic outpourings of concern may have little to no basis in reality, ‘it typically leaves in its wake long-standing institutional changes that continue to affect adversely the marginalized’ [40], p. 12. Negative representations lead to further policing, profiling and scrutiny of African diaspora communities, and solidify the synonymy of Africanness and blackness with negative racial codes and cultural meanings [21]. This was clearly demonstrated through efforts to contain coronavirus outbreaks in Australia which resulted in the overpolicing and stigmatization of African migrants in particular [44,45].

Studies show that racial discrimination is a social determinant to health inequities among racially minoritised communities [46]. William Smith describes the significant negative physiological, psychological and health impacts of the ongoing onslaught of racism as racial battle fatigue (RBF) [47]. Speaking about the impact of overpolicing, Sudanese youth in Melbourne for example have reflected on consistently feeling unable to protect themselves from negative media and feeling unworthy of being included in mainstream Australia [12]. They emphasized the burden of the emotional toll of these experiences that wore them down, drained them and impacted their health and wellbeing [43].

For Africans in Australia therefore, their Black embodiment and the associated racialized scripts they encounter can lead to racial battle fatigue being woven through their daily life. As documented by Mapedzahama and Kwansah-Aidoo [7], blackness can be experienced as a ‘burden’ that must be carried and constantly negotiated in the Australian social fabric, which consequently amplifies experiences of strangerhood. A key study by Ferdinand, Paradies, and Kelaher [46], p. 7 demonstrated that experiences of racism contributed to poor mental health and in some cases negative physical health outcomes which “highlight the need for interventions to protect the mental and physical health of racial and ethnic minority communities”.

### 5.2. Perpetual Strangerhood

While Australians of African descent may live in Australia as law-abiding and productive citizens, the ongoing scrutiny, questioning, and unending construction of their status within the nation-state and within ‘local communities’ may nevertheless position them as ‘perpetual strangers’. When Black Africans migrate to Western and settler-colonial societies, they became ‘objects of curiosity’ for the white gaze [4]. They are marked as ‘visibly different’; they are ‘recognizable as different from the white, Western-clad, and English-speaking majority in various ways, including phenotype, attire, accent, or a combination of these ‘visibilities’ [48]. Individuals from a wide variety of cultural, ethnic, and national backgrounds are relegated to singular categories such as ‘African’ or ‘Black’—labels by which they may never have previously identified with. So, in addition to becoming hyper-visible as a body, Black African Immigrants can also experience the paradoxical opposite; that of becoming invisible. Both can lead to the imposition of ‘strangerhood’ and denial of belonging. As [49], p. 2 suggest, in Australia, whiteness is the norm, ‘the standard against which differences, or deviations from that norm, are measured, valued, and often demeaned’.

Conditionality of belonging has been an increasingly prominent feature of the globalized world in the past twenty years, as securitization agendas construct ‘foreigners’ or ‘strangers’ not only as ‘a threat to the cohesion of the political and cultural community but also as potential terrorists’ [19]. As Yuval-Davis [19], p. 213 emphasizes, ‘the politics of belonging has come to occupy the heart of the political agenda almost everywhere on the globe, even when reified assumptions about “the clash of civilizations” are not necessarily applied’. African migrants confront a complex postcolonial conundrum of needing to navigate anti-Black rhetorics of ‘belonging’ based on ‘borders that are inherently porous, of colonial origin, and paradoxically symbolic of sovereignty’ [50]. Proprietary belonging by Australians of white settler-colonial heritage can also be ‘reflected in the traditional vigilance and policing of boundaries between “locals” and “Others”’ [26].

One of the most ubiquitous ways in which this boundary is policed in Australia is through the racialized question ‘where are you from?’ [1]. As Kwansah-Aidoo and Mapedzahama, [28] have noted, this question, repeatedly directed towards Black Africans and other visibly different immigrants ‘imaginatively dislocates them from “here” and makes them strangers in a familiar land’. They state: 

*The question symbolically deports [the interrogated person] back to the faraway places “where they are from” … they are not “authentic Australians” because their visible difference (attributable to their skin colour) impedes their inclusion in the imagined Australian nation. Yet they are not authentic foreigners because, apart from having Australian citizenship, some of them have been here too long to be bona fide foreigners. Thus, while this question may enable the questioner to ‘re/locate’ the questioned to some distant geopolitical location, it also imaginatively dislocates them from ‘here’ and makes them strangers in a familiar land*.[28], p. 108

Udah and Singh argue that this question reminds the Black African migrant that they are not recognized as ‘locals…but are considered as strangers born in some far-away place’ [1]. Nyuon articulates the implications of such symbolic deportation by wondering ‘what it would feel like to feel Australian but happen to be black?’ She asks, ‘how do you hold on to a sense of belonging when it is so often assaulted by racism?’ [51].

A dominant theme of the literature on African diasporic experiences in Australia is the way that visible differences conjure racialized coded meanings that can lead to high levels of unemployment, underemployment, and a loss of post-migration occupational status [52]. The labour market structurally excludes Africans from the job market through policing of ‘English proficiency, non-recognition of overseas qualifications and skills and a lack of Australian experience’, which contributes to a downward socioeconomic spiral [52,53]. The preference for Western expertise assumes those who are not from Western countries are ‘strangers’ to western knowledge, regardless of their training and experience. This means that even when they acquire work, Africans are more likely to experience subtle, persistent and normalized racial microagressions or biased assumptions about their ability to do the job as well as White Australians [53]. Mapedzahama locates these experiences as examples of ways that black bodies in white spaces are ‘speaking a language of [their] own’, one that works to essentialize, and homogenize them [4].

Black people can face significant backlash when they speak publicly about their racialized experiences in Australia, with increased severity experienced by those with more marginalised intersecting social identities such as women from African migrant backgrounds [27]. In a recent example, Nyuon, a Sudanese-Australian lawyer based in Melbourne has publicly documented the significant impacts of cyber-bullying she has experienced due to her outspokenness about racism [54]. As African migrants are expected to perform perpetual gratitude towards Australia for ‘letting them in’, representatives of dominant social groups, under the guise of ‘free speech’, are licensed to debate the existence and harmlessness of racism which minimizes the racial trauma fueled by the racial violence and microagressions that Black people experience everyday [27]. Always required to be cognizant of the hyper-visibility of their skin colour, Black African migrants can come to terms with the difficult and transcendental journeys of migration by embracing their blackness, resenting it, or feeling separated from it. This can be a liberating experience, but it can also foster a deep sense of exclusion, and the impression that it is necessary to be white to be fully accepted and to exist with dignity in Australia.

## 6. Impact of Racialization on Health

Health and social research support the evidence that racism is a social determinant of health and that it contributes to a disproportionate health inequality and poor access to services for racialised populations [55,56,57]. As Marmot argues, “inequities in health arise from inequities in society” [58], p. 512. Taking a critical race theory (CRT) perspective, we argue that processes of “racial subordination, prejudice, and inequity” produce experiences of exclusion and fractured belonging and have a significant human cost in terms of mental wellbeing [55,59]. A CRT perspective explores how complex processes of racialization are enacted in health care and in health seeking behaviours [60]. Within an Australian context, CRT focuses particularly on the critical understandings of health and wellbeing for Black people, particularly Aboriginal and Torres Strait Islander peoples, while also attempting to de-center whiteness as the standard by which experiences of health and wellbeing are measured [49]. Within this theorization, whiteness is not seen merely as a skin colour but rather a social process of racial hierarchical structuring where markers of value, universality and social capital are consciously or unconsciously bequeathed upon white people. Whiteness consequently dominates other ways of knowing, being and experiencing the world. Through this assumed universality, the western biomedical model of health care amplifies white- and Euro-centric health practices which can unwittingly perpetuate racialised pathologization of Black people. Without critically examining how race and colour-blind approaches to health care impact minoritized peoples, the health care system itself becomes complicit in perpetuating racism [61].

As Durey and Thompson argue, ‘people often view racism solely as referring to interpersonal relations, where a person is treated unfairly … because of race. However, racism that exists systemically and institutionally, where the production, control and access to resources operates to advantage selected racial/cultural groups and disadvantage others, is more insidious’ [49], p. 3. Continually highlighting the health gaps and inequalities between Black people in comparison to white people without a critical theorization of the colonial, historical, social, cultural, and political inequities driving those “gaps” perpetuates rhetorics that pathologize blackness [62], which in and of itself, is form of racial gaslighting. As defined in the National Aboriginal Health Strategy, health for Indigenous peoples (and other colonized Black peoples), health is:

*Not just the physical wellbeing of the individual, but the social, emotional and cultural wellbeing of the whole community … [it is] a matter of dignity, of community self-esteem and of justice. It is not merely a matter of the provision of doctors, hospitals, medicines or the absence of disease and incapacity*.[56], p. 26

Addressing racism and discrimination is a public health imperative. In their research, Ferdinand, Paradies, and Kelaher suggest that ‘preventing racial discrimination will be a more constructive approach to protecting the health of racial and ethnic minority communities than relying on the use of appropriate response mechanisms after a racist incident has occurred’ [46], p. 12. As such, transforming cultures of care must start with the acknowledgement that different forms of historical, structural as well as interpersonal oppression combine and contribute to the poor health and chronic physical and psychological ailments that are suffered by racialised and colonized peoples in Australia and globally [49,57]. 

## 7. Conclusions

The emerging literature on Black African migrant experiences in Australia highlights not only the precarities of conditional belonging, constant boundary-work and the resulting experiences of battle fatigue that they must contend with on a variety of systemic and quotidian levels, but also the variety of strategies of resilience that are employed in response to these challenging circumstances. Among the solutions highlighted in the literature are finding belonging and solidarity within tightly-knit migrant communities [43] to enable enough agency to exercise rights of governmentality over space, and developing a ‘negotiated’ or ‘hyphenated’ borderlands identities ‘so that both cultures can become a part of how they inhabit space in Australia’ [63]. As we have also highlighted in this paper, there are significant implications for public health policy and practice in understanding and combating the impacts of racism in order to support the wellbeing and health of Black African migrants. As ongoing conversations and debates about migration are complicated by the rising influence and amplification of nationalist discourses in the context of international border closures and rising social anxiety, it becomes imperative that the stories and experiences of Africans in Australia be effectively shared and documented with a dignified sociological nuance, and that the significance of anti-racism within Australian public health discourses be substantially amplified.

Navigating the formation of novel and resilient diasporic identities is a key theme in the literature regarding African migration to Australia. Moral panics and the construction of Black African ‘strangerhood’ raise particular challenges (and contradictions) for traversing the fallout from reified and homogenized black/African migrant/outsider labels and representations [7]. Because of such experiences, blackness can be carried as a burden within a settler-colonial society such as Australia. As African migrants are among the most ‘visible’ social groups in Australia in terms of phenotypical differences, the significant problems relating to their marginalization and minoritization extend beyond poor physical, psychological and economic outcomes. Given that this is an extremely salient aspect of migrant subjectivity, we see a need to further research investigating the nuances of new diasporic identities, and how they metamorphosize through different environments and experiences. More so, it is crucial to probe how Black Africans develop resilience in countering the impacts of hypervisibility and scrutinization and of being rendered invisible.

We conclude this paper by considering the words of Falkof who emphasized that moral panics have ‘ideological motives, they are stories that we tell ourselves and each other to help us make sense of insecurity and social change’ [41]. This means there are ways that new stories about what it means to be Australian can be told, other than the continual recycling of racialized moral panics of one form or another. To accommodate the diversity of Australian culture and the subjectivity of each experience, a significant re-imagining of the Australian community is required, one that can own up to the past and current realities of racism and its impacts and find a place for those historically constructed as ‘strangers’ within the nation. This is a challenging proposition, but this nascent potential is within our grasp.

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
