# Peer review of "Boundaries of Belonging: Theorizing Black African Migrant Experiences in Australia"

_ijerph, 2020, doi:10.3390/ijerph18010038_

Round 1

Reviewer 1 Report

I like this article a lot. It is theoretically well grounded and offers a unique perspective into the issues of blackness and immigration in Australia. I reckon the piece has the potential to change the way research is done on this topic in Australia if the author could further theoretically interrogate what he/she beautifully express in the following terms: “reductionist arguments that singularize the experiences of diasporic Africans”. I would like to encourage the author to drill deeper into this perceptive observation because there is a lot to be mined out of this observation. For instance, the practice of singularising experience of diasporic Africans is partly shaped by the Refugee Studies framework which compels researchers to singularise group experience. This practice is also influenced by liberal white framing of black experience. Liberal white frames are misinformed about the global history of blackness.

A minor issues which I think maybe reflects my lack of familiarity with the reference style used by the author is that there are certain sentences that I think deserve to be referenced which the author has not referenced. I am of the view that as researchers who research and write about settler societies in a way that mainstream white societies find offensive almost, it is highly imperative that we are extra vigilant in our writing and make sure we do not make ourselves unnecessarily vulnerable. For instance, I feel that the author should have referenced the following sentence:

“In this paper we use ‘Black and Blackness’ to refer only to ‘Afro-Blackness’ or the blackness that is experientially embodied by people identified as ‘Black Africans’. We do this while also acknowledging that not all Black people are of African descent, and not all Africans are Black. While we are careful not to homogenize the experiences of Black Africans in Australia, we acknowledge that bodies that are visibly marked as ‘Black’ and ‘African’ can share some similarities in their experience.”

From the above quote I can recognise Achille Mbembe’s insight, as well as things I have said in the past, and what has been said by others in different contexts. Again, maybe this is a minor issue, maybe the author does reference this sentence, and maybe I missed it. Regardless, I would like to bring this to the attention of the author.

Author Response

Comment 1

I like this article a lot. It is theoretically well grounded and offers a unique perspective into the issues of blackness and immigration in Australia. I reckon the piece has the potential to change the way research is done on this topic in Australia if the author could further theoretically interrogate what he/she beautifully express in the following terms: “reductionist arguments that singularize the experiences of diasporic Africans”. I would like to encourage the author to drill deeper into this perceptive observation because there is a lot to be mined out of this observation.

Response: 

Thankyou for this generous and encouraging feedback. We have taken this advice and sought to develop our line of theorisation more robustly and to develop key observations from the paper within the constraints of the word limit .

Comment 2

A minor issues which I think maybe reflects my lack of familiarity with the reference style used by the author is that there are certain sentences that I think deserve to be referenced which the author has not referenced. I am of the view that as researchers who research and write about settler societies in a way that mainstream white societies find offensive almost, it is highly imperative that we are extra vigilant in our writing and make sure we do not make ourselves unnecessarily vulnerable. For example, referencing the following:

“In this paper we use ‘Black and Blackness’ to refer only to ‘Afro-Blackness’ or the blackness that is experientially embodied by people identified as ‘Black Africans’. We do this while also acknowledging that not all Black people are of African descent, and not all Africans are Black. While we are careful not to homogenize the experiences of Black Africans in Australia, we acknowledge that bodies that are visibly marked as ‘Black’ and ‘African’ can share some similarities in their experience.”

From the above quote I can recognise Achille Mbembe’s insight, as well as things I have said in the past, and what has been said by others in different contexts. Again, maybe this is a minor issue, maybe the author does reference this sentence, and maybe I missed it. Regardless, I would like to bring this to the attention of the author.

Response:

Thank you for this suggestion. We are mindful of the exceptional standards of scholarship required for critical race studies, and appreciate these suggestions regarding referencing. We have added a reference for the nominated point, and reviewed the rest of the paper and added some further references at key points with your suggestion in mind as well 

Reviewer 2 Report

Leanne Weber's direct quote, sentences 122-125, should include the page number for the source.

Page number for the source of the direct quote by Kwansah-Aidoo and Mapedzahama (Sentences 339-343) needs to be included.

The direct reference to Nyadol Nyuon (sentence 363-364) raises confidentiality concerns. Perhaps she should be quoted without revealing her name.

The discussion in paragraph 302-312 will be enriched by the following study:

Mental Health Impacts of Racial Discrimination in Australian Culturally and Linguistically Diverse Communities: a cross-sectional survey by Angeline S Ferdinand, Yin Paradies, and Mragaret Kelaher

BMC Public Health.2015;15:401

Author Response

Comment 1:

Leanne Weber's direct quote, sentences 122-125, should include the page number for the source.

Response: Thank you for noting this omission, the page numbers have now been added.

Comment 2: Page number for the source of the direct quote by Kwansah-Aidoo and Mapedzahama (Sentences 339-343) needs to be included.

Response: Thank you for noting this omission, the page numbers have now been added .

Comment 3: The direct reference to Nyadol Nyuon (sentence 363-364) raises confidentiality concerns. Perhaps she should be quoted without revealing her name.

Response: Thank you for highlighting this concern. In our paper we cited Nyadol Nyuon’s published work where she utilised her own name, in places such as The Guardian and which are in the public domain. She was not a participant in research with us and did not provide the comments directly to us in confidence.

Comment 4: The discussion in paragraph 302-312 will be enriched by the following study:

Mental Health Impacts of Racial Discrimination in Australian Culturally and Linguistically Diverse Communities: a cross-sectional survey by Angeline S Ferdinand, Yin Paradies, and Mragaret Kelaher BMC Public Health.2015;15:401

Response: Thank you for the recommendation, this is indeed a helpful source which we have now integrated into the paper .

Reviewer 3 Report

I enjoyed reading the work.  The work provides a rich insight into African migrant experience in Australia.  My main concern with this paper is the journal it has been submitted to.  "International Journal of Environmental Research and Public Health (IJERPH) (ISSN 1660-4601) is a peer-reviewed scientific journal that publishes original articles, critical reviews, research notes, and short communications in the interdisciplinary area of environmental health sciences and public health".  The focus of this journal is on public health.  However, there is no information provided in the article on health or on public health.  This will be a great paper to publish in a social science journal but I doubt the appropriateness of this paper in this specific journal. An alternative is to link discussion about community belonging to health so that this paper address a health issue.

Author Response

Reviewer's comments

I enjoyed reading the work. The work provides a rich insight into African migrant experience in Australia. My main concern with this paper is the journal it has been submitted to. "International Journal of Environmental Research and Public Health (IJERPH) (ISSN 1660-4601) is a peer-reviewed scientific journal that publishes original articles, critical reviews, research notes, and short communications in the interdisciplinary area of environmental health sciences and public health". The focus of this journal is on public health. However, there is no information provided in the article on health or on public health. This will be a great paper to publish in a social science journal but I doubt the appropriateness of this paper in this specific journal. An alternative is to link discussion about community belonging to health so that this paper address a health issue. 

Response:

Thankyou for this suggestion. The Editor broadly agreed with your concerns, however wished us to revise the paper in order to bridge the gap between social sciences scholarship and the core public health readership for this journal. They supported ongoing pathway to publication in this journal with appropriate revisions addressing your concerns.

Reviewer 4 Report

The authors assess a subject of growing interest, as the number of African immigrants in Australia are day by day increasing and their integration processes need -and deserves- more space in the research literature. 

The theme of the paper is described in details, assessing subjects such as "belonging" and "blackness". Also, the authors provide fair bibliographic references.

However, some details are missing and would provide an improvement of the paper. 
In particular, a deeper contextualisation paragraph is needed. The article should contain an exhaustive section detailing the immigrant population of Australia, in order to contextualise -also by providing "numbers"- the presence of African immigrant population.

Author Response

The authors assess a subject of growing interest, as the number of African immigrants in Australia are day by day increasing and their integration processes need - and deserves - more space in the research literature. The theme of the paper is described in details, assessing subjects such as "belonging" and "blackness". Also, the authors provide fair bibliographic references. However, some details are missing and would provide an improvement of the paper. In particular, a deeper contextualisation paragraph is needed. The article should contain an exhaustive section detailing the immigrant population of Australia, in order to contextualise -also by providing "numbers"- the presence of African immigrant population.

Response

Thank you for this suggestion, we have reviewed the contextualising information within the paper and made further additions as per your suggestions